# Aging of the Nigrostriatal Tract in the Human Brain: A Diffusion Tensor Imaging Study

**DOI:** 10.3390/medicina57090994

**Published:** 2021-09-20

**Authors:** Jeong-Pyo Seo, Dong-Kyun Koo

**Affiliations:** Department of Physical Therapy, College of Health Sciences, Dankook University, Cheonan 31116, Korea; raphael0905@hanmail.net

**Keywords:** nigrostriatal tract, diffusion tensor tractography, aging, Parkinson’s disease, dopaminergic pathway

## Abstract

*Background and Objectives*: The loss of dopamine neurons in the nigrostriatal tract (NST) is one of the main pathological features of Parkinson’s disease (PD), and degeneration of the NST leads to the motor symptoms observed in PD, which include hypokinesia, tremors, rigidity, and postural imbalance. In this study, we used diffusion tensor tractography (DTT) to investigate the aging of the NST in normal human subjects to elucidate human brain structures. *Materials and Methods*: Fifty-nine healthy subjects were recruited for this study and allocated to three groups, that is, a 20 to ≤39 year old group (the young group), a 40 to ≤59 year old group (the middle-aged group), and a ≥60 year old group (the old group). DTT scanning was performed, and NSTs were reconstructed using the probabilistic tractography method. NSTs were defined by selecting fibers passing through seed and target regions of interest placed on the substantia nigra and the striatum. *Results*: A significant negative correlation was observed between age and fractional anisotropy and tract volume (TV) of the NST. Mean TV values of the NST were significantly lower in the old group than in the young and middle-aged groups (*p* < 0.05). The TV values of the NST were significantly reduced with age for men and women (*p* < 0.05). *Conclusion*: We found that aging of the NST began in the 3rd decile and progressed steadily throughout life until old age, when it exhibited significant degeneration. We suspect these results are related to the correlation between the incidence of PD and age.

## 1. Introduction

Ageing is an unavoidable process in an organism’s lifespan, accompanied with physical decline and an increased risk of disease and mortality [1]. The rate of deterioration with age varies between species, individuals, and tissues [2]. Neurodegenerative diseases and their accompanying cognitive deficiencies are prevalent in older populations, compromising their healthy lifespan and quality of life. When it comes to the numerous risk factors that have been related to neurodegeneration, ageing biomarkers are by far the most influential [3]. It is possible that brain aging is a scale of neurodegeneration progression, and that human genetic and environmental factors have a role in the onset and progress of neurodegenerative disease [3,4].

The nigrostriatal tract (NST) projects from the substantia nigra pars compacta to the striatum and is one of the major bilateral dopaminergic pathways that links the brainstem and the striatum [4,5,6]. The NST is one of the four major dopamine pathways in the brain and is a component of the basal ganglia motor loop which critically underlies the generation of movements [7]. Degeneration of the NST is presumed to be the cause of the motor symptoms observed in PD, which include hypokinesia, tremors, rigidity, and postural imbalance [4,8].

As detailed knowledge regarding the normal aging of specific brain structures aids the development of strategies aimed at preventing or delaying aging, many diffusion tensor tractography (DTT) studies have been performed because this modern technique enables three-dimensional qualitative visualization and analysis of neural structures [9,10,11,12,13,14]. However, few such studies have focused on the NST in patients with PD [6,15,16,17], and no study has yet addressed the effect of aging on the NST. In the current study, we examined aging of the NST in normal human subjects using DTT.

## 2. Methods

### 2.1. Subjects

Fifty-nine right-handed, healthy subjects (males: 35, females: 24, mean age: 46.5 years, range: 20–78 years) with no previous history of a psychiatric, neurological, or physical illness and no brain lesion according to conventional MRI (T1-weighted, T2-weighted, Fluid attenuated inversion recovery [FLAIR] or T2-weighted gradient recall echo [GRE] images), as confirmed by a neuroradiologist, were enrolled in the present study. Subjects were divided into three groups by age decade (young group: 20–39 years, middle-aged group: 40–59 years, old group: 60–79 years). All subjects provided written informed consent prior to study commencement, and the study protocol was approved by the Institutional Review Board of Yeongnam University Hospital.

### 2.2. Diffusion Tensor Imaging

Diffusion tensor imaging data were acquired using a Synergy-L SENSE head coil on a 1.5T Gyroscan Intera system (Philips, Best, The Netherlands) equipped with single-shot echo-planar imaging. For each of the 32 non-collinear diffusion sensitizing gradients, 67 contiguous slices were acquired parallel to the anterior commissure–posterior commissure line. Imaging parameters were as follows: acquisition matrix = 96 × 96, reconstructed matrix = 192 × 192 matrix, field of view = 240 × 240 mm^2^, TR = 10,398 ms, TE = 72 ms, parallel imaging reduction factor (SENSE factor) = 2, EPI factor = 59 and b = 1000 s/mm^2^, NEX = 1, slice gap = 0, and slice thickness 2.5 mm.

### 2.3. Diffusion Tensor Tractography

The Oxford Centre for Functional Magnetic Resonance Imaging of the Brain (FMRIB) Software Library was used to analyze diffusion-weighted imaging data (www.fmrib.ox.ac.uk/fsl (accessed on 20 April 2020)). Affine multi-scale two-dimensional registration was used to correct for head motion effects and image distortions due to eddy currents. A probabilistic tractography method based on a multi-fiber model was used for fiber tracking and was applied utilizing tractography routines implemented in FMRIB Diffusion (step length 0.5 mm, 5000 streamline samples, curvature threshold = 0.2) [18,19,20,21].

NSTs were delineated by selecting fibers that passed through seed and target regions of interest (ROI). For each participant, a seed ROI was located at the substantia nigra on the fractional anisotropy (FA) map at midbrain, and a target ROI was placed on the striatum on the FA map (Figure 1). We set the ROIs as known brain anatomy manually [17]. Of the 5000 samples generated from each seed voxel, results for each contact included visualized threshold and weightings of tract probability for a minimum of one streamline through each analyzed voxel. FA and tract volume (TV) of the NST were measured.

### 2.4. Statistical Analysis

SPSS software (v.15.0; SPSS, Chicago, IL, USA) was used for data analysis. Pearson’s correlation analysis was used to assess the significances of correlations between these two DTT parameters and age. One-way analysis of variance (ANOVA) with the LSD post-hoc test was used to determine the significances of differences for each DTT parameter (FA and TV) between age groups by sex. In addition, an independent t-test was performed to investigate differences in DTT parameters according to sex between each group. Statistical significance was accepted for *p* values < 0.05. Measured NST values are presented as means (±standard deviations).

## 3. Results

Correlations between age and the two DTT parameters are summarized in Figure 2. Age did not show a significant correlation with the mean FA value of NST (*r* = −0.231) (*p* > 0.05). However, the mean TV value of NST showed a significant negative correlation with age (*r* = −0.484) (Pearson’s correlation, *p* < 0.05) [22]. Table 1 shows the comparison of the mean values of the NST parameters in the three groups. There was no significant difference in the mean FA value of NST between the three groups (*p* > 0.05). However, the mean TV values of NST showed a significant difference between the three groups (*p* < 0.05) (Table 1). In post-hoc analysis, the mean FA value of NST did not show a significant difference between all groups (*p* > 0.05). In addition, there was no significant difference in the mean TV values of NST between middle and young groups (*p* > 0.05). However, post-hoc tests revealed that the mean TV values of NST in old group were significantly different from those of middle and young groups (*p* < 0.05) (Table 1).

Table 2 shows the comparison of the mean values of the NST parameters in the three groups by sex. There was no significant difference in the mean FA value of NST between the three groups in any sex (*p* > 0.05). However, the mean TV value of NST showed a significant difference between the three groups in the females and males (*p* < 0.05) (Table 2). Comparing DTT parameters between males and females, there were no significant differences in the mean FA and TV values of NST between sex in all groups (young, middle, and old groups) (*p* > 0.05).

## 4. Discussion

In the current study, we investigated the aging of the NST in the normal human brain with diffusion tensor imaging. According to our findings: (1) The mean FA and TV values of the NST showed an overall decrease with age, (2) the mean TV values of the NST were lower in the old group than in the middle-aged or young groups, and (3) the TV values of the NST were smaller for old men than for the middle-aged or young men and smaller for old women than young women. FA, which describes the degree of diffusion anisotropy, is a prominent scalar quantity. It indicates the degree of directionality and the integrity of white matter microstructures such as axons, myelin, and microtubules [23,24]. For each white matter pathway, TV was computed by summing the number of voxels with at least one streamline and multiplying by the voxel volume. TV values reflect the total number of fibers in a neural tract [25]. Age-related declines in FA values and reductions in the numbers of myelinated fibers in a neural tract can reduce TV values [10,24,26]. Our results indicate that age-related NST degeneration commences at NST ends in the 3rd decile and then progresses steadily at a near-constant rate until the 8th decile, when degeneration accelerates.

Previous studies on the aging of the dopaminergic pathway in the normal human brain have demonstrated that age-related changes (e.g., dopaminergic neuronal loss) in the dopaminergic pathways of PD patients occur at an almost constant rate throughout life [27,28,29]. Furthermore, it has been reported that TV value changes from middle to old age in men and women differ and that TV values reduce from middle to old age in men and women, which may reflect demyelination and a decrease in the number of myelin fibers [30]. We believe that the present study supports clinical and experimental evidence that males are more susceptible to PD than females [31]. Moreover, the volume of the crus of fornix and the volume of the ventromedial prefrontal white matter gradually decreased with age. Our results support that the majority of white matter tracts follow the same developmental trajectory with aging as the global white matter volume of the brain [32]. However, previous neurobehavioral studies have not involved evaluations of the NST. Although many DTI studies have been conducted on neural tract aging in the human brain [10,33], to the best of our knowledge, this is the first study to investigate the effect of aging on the NST in the normal human brain.

In summary, we found that the aging of the NST commenced in the 3rd decile and progressed steadily at a near-continuous rate throughout life until the 8th decile when the rate of degeneration of the NST increased significantly. We suspect these results are related to the correlation between PD and age in normal subjects. The limitations of the present study include a lack of data on individuals older than 80 and of clinical data related to NST function. We also admit the possibility that factors other than age may have contributed to our findings. In addition, the present study did not consider information on mutations in genes responsible for PD or family histories of PD or other neurodegenerative disorders or Lewy body pathology. This study was a retrospective study conducted on normal people and did not enroll Parkinson’s disease patients. It is necessary to investigate additional Parkinson’s disease patients in further studies. The limitations of DTI should be considered. Due to the crossing fiber or partial volume effect, DTI may underestimate fiber tracts, and it is difficult to represent all fibers, particularly small fibers [34,35]. Therefore, we suggest that further prospective studies be conducted on individuals older than 80 and that neurobehavioral data be collected to enable the adequate assessment of NST function.

## Figures and Tables

**Figure 1 medicina-57-00994-f001:**
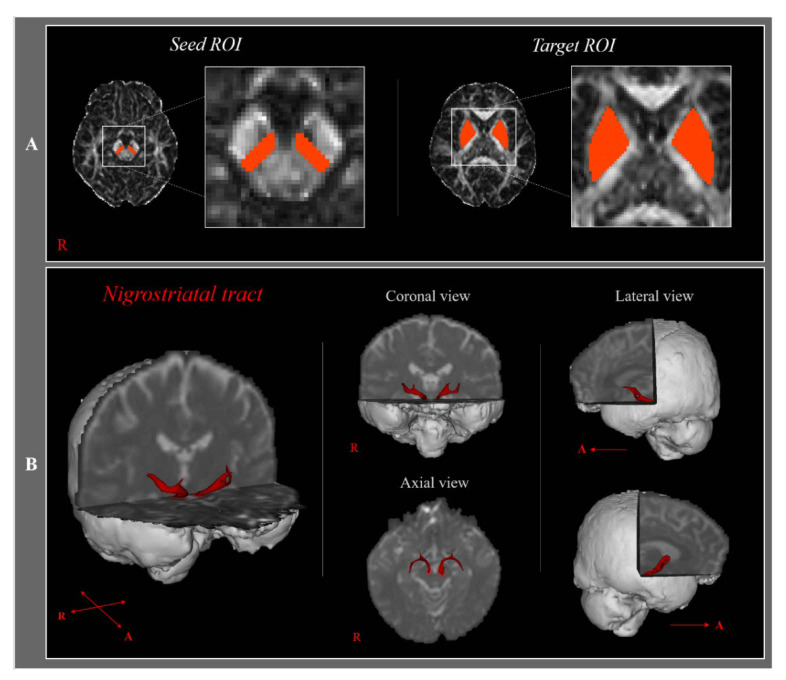
(**A**)—The region of interest (ROI) and diffusion tensor tractography for the nigrostriatal tract. The first ROI was placed on the substantia nigra, and the second ROI was placed on the striatum. (**B**)—In both hemispheres, the NST is being reconstructed. T2-weighted brain magnetic resonance images showing a representative subject (20-year-old female); R: right; A: anterior.

**Figure 2 medicina-57-00994-f002:**
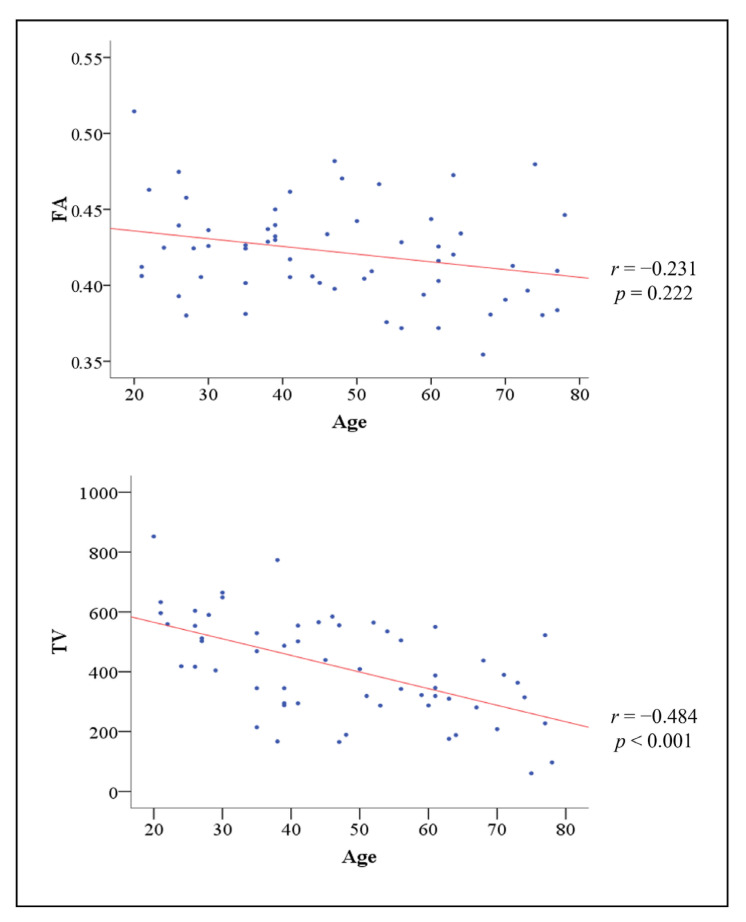
Correlations between the age of all subjects and the two DTT parameters.

**Table 1 medicina-57-00994-t001:** Mean values of diffusion tensor tractography parameters of the nigrostriatal tracts.

		Young	Middle	Old	*p*-Value
Nigrostriatal tract	FA	0.429 (±0.03)	0.422 (±0.03)	0.412 (±0.03)	0.241
	TV	494.354 (±168.24)	419.647 (±139.31)	303.611 (±131.43)	0.001 *
		Young vs. Middle	Middle vs. Old	Young vs. Old	
Post-hoc *p*-value	FA	0.447	0.395	0.093	
	TV	0.121	<0.001 *	0.026 *	

FA: fractional anisotropy, TV: tract volume (voxels); Values indicate mean (±standard deviation); * *p* < 0.05 ANOVA.

**Table 2 medicina-57-00994-t002:** Mean values of diffusion tensor tractography parameters of the nigrostriatal tracts according to sex.

			Young	Middle	Old	ANOVA *p*-Value
Nigrostriatal tract	Males	FA	0.428 (±0.03)	0.428 (±0.03)	0.409 (±0.03)	0.256
		TV	445.885 (±151.56)	423.900 (±143.12)	285.583 (±140.28)	0.022 *
	Females	FA	0.432 (±0.04)	0.413 (±0.03)	0.419 (±0.04)	0.518
		TV	551.636 (±175.71)	413.571 (±144.77)	339.667 (±114.46)	0.032 *

FA: fractional anisotropy, TV: tract volume (voxels); Values indicate mean (±standard deviation); * *p* < 0.05 ANOVA.

## Data Availability

The data presented in this study are available on request from the corresponding author.

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
