# Peer review of "Aging of the Nigrostriatal Tract in the Human Brain: A Diffusion Tensor Imaging Study"

_medicina, 2021, doi:10.3390/medicina57090994_

Round 1

Reviewer 1 Report

Seo and Koo report a study examining aging of the nigrostriatal tract (NST) via probabilistic diffusion tensor tractography in 59 human subjects separated into 3 age cohorts. They found a weak / moderate negative correlation between age and fractional anisotropy / tract volume.

While I believe that there is some significance to the content, I have some issues that should be reflected by the authors:

Even though, the study provides only data of healthy ageing subjects, a quite substantial part of the manuscript focuses on Parkinson’s Disease (PD). The implication to use this normative data in studies on PD may be mentioned once or twice (e.g. linking to a review like Zhang & Burock 2020, Front.Neurol 10.3389/fneur.2020.532993), but further elaboration suggests conclusions that cannot be drawn and some information given just seem to be wrong. For example, second sentence of the introduction: GABA interneuron turnover might play a role in the pathophysiology of PD, but being the primary cause of clinical symptoms is far-fetched. The major mechanism is the gradual loss of dopaminergic cells in the substantia nigra, the underlying primary cause not known or at least debatable. Age is seen (by most experts on the field) more as a risk and maybe accelerating factor than as a cause.

On the other hand, a discussion of the findings in the context of ageing of other brain regions in the brain (or the whole brain) is mostly missing. Do the findings align with brain ageing studies of other regions (e.g. Michielse et al. 2010, NeuroImage, 10.1016/j.neuroimage.2010.05.019) and/or do they support theories like the “last-in-first-out” hypothesis (e.g. in Davis et al. 2009, NeuroImage 10.1016/j.neuroimage.2009.01.068)?  

Some limitation issues (and how they were considered) should be discussed:

-Partial voluming effects (due to CSF contamination, CSF pulsation, age-related increases in extracellular spaces and age-related atrophy).

-Singularity Problem (crossing fibres).

-Representation of indicative rather than actual fibres (Do they find actual fibre loss or only loss of ease of tracking?).

-Potential inclusion of subjects with sub-clinical Lewy body pathology in the old group (risk of developing PD over 60 within the next ten years is approx. 1:100 (Hirtz et al 2007 Neurology, Driver et al 2009 Neurology) and increases further with age; risk of subclinical Lewy body pathology is even 10fold higher (Fearnley & Lees 1991 Brain).) This might have influenced results of the old group (and account for all significance?).

Since the definition of ROI is very critical for results, a bit more details should be given (any control mechanisms, standardized procedure?).

Why did the authors choose FA and TV as outcome measures, and refrained from diffusivity measures.

Minors:

p.1 line 11: spelling of tractography

p.1 lines 12-14: “Fifty-nine healthy subjects were recruited for this study and allocated to three groups, that is, a 20 to hy subjects were recruited for this studym to ≤ 59 year old group (the middle-aged group, and a ≥ 60 year old group (the old group).” -> not understandable

p3 line 96: spelling of presented

figure 2: really THREE DTT parameters? Missing of unit definition (years, voxel). Blue dots represent single subjects?

p5 line 134 spelling of TV value changes

Reviewer 2 Report

The present article is poorly written. Just reading the abstract gives an idea of the whole. Above all, this article lacks novelty as many good articles have been published reporting the same findings. 

I have substantial comments that the authors should address:

  1. Why only healthy participants? It might be very informative to compare with PD patients. Moreover, is there sex-dependent differences as no test was performed. The author did not take multiple comparisons into consideration?

      2. FA and TV should be defined clearly to be more educational. 

      3. The authors stated that the NST FA and TV values showed an                    overall decrease with age, while the results are not significant. 

      4. It is not clear how the given P-values are calculated and which           columns are compared. 

Round 2

Reviewer 2 Report

No further comments. The authors adequately responded to my comments.